# Surface-Enhanced Raman Scattering Studies of Au-Ag Bimetallic Nanoparticles with a Tunable Surface Plasmon Resonance Wavelength Synthesized by Picosecond Laser Irradiation

P. Babuji [1], Md Abu Taher [2], Mudasir H. Dar [3], D. Narayana Rao [4], P. Gopala Krishna [1] and V. Saikiran [1,*]

[1] Department of Physics, School of Sciences, GITAM Deemed to Be University, Visakhapatnam 530045, India; bputiki@gitam.in (P.B.); gpodagat@gitam.edu (P.G.K.)

[2] Department of Electrical Engineering, Indian Institute of Technology Hyderabad, Hyderabad 502285, India; taherphys20@gmail.com

[3] Department of Physics, Govt. Degree College, Beerwah, Budgam, Jammu and Kashmir 193411, India; mhdar09@gmail.com

[4] School of Physics, University of Hyderabad, Hyderabad 500046, India; dnr.laserlab@gmail.com

\* Correspondence: svadaval@gitam.edu

**Abstract:** Here, we present a simple and green method of preparing Au-Ag bimetallic nanoparticles (NPs) with a tunable surface plasmon resonance (SPR) wavelength by using picosecond laser irradiation. Au-Ag alloy NPs have been produced by irradiating the solutions containing respective metallic salts in a polyvinyl alcohol (PVA) matrix using a picosecond laser in a single-step process. The SPR wavelength of the Au-Ag bimetallic NPs is observed to be shifted/changed with the Au-Ag concentration and the laser irradiation parameters. The Au-Ag NPs embedded in the PVA matrix are advantageous for Surface-Enhanced Raman scattering (SERS) applications. The estimated enhancement factors (EFs) were observed to vary as a function of conditions of the Au-Ag bimetallic alloy NPs synthesis and also on the concentration of Au at a fixed input fluence of irradiation. The SERS active platforms of Au-Ag bimetallic NPs showed EFs as high as of the order of $10^8$ for Crystal Violet (CV) dye samples at nano molar concentrations. The present study demonstrates a simple, single-step, and green method that fabricates Au-Ag alloy-based nanocomposites suitable for SERS investigations with significantly higher orders of EFs.

**Keywords:** Au-Ag bimetallic alloy nanoparticles; laser irradiation; SERS; surface plasmon resonance

## 1. Introduction

The giant enhancement in the Raman scattering signal of a molecule in the presence of metal nanoparticles (NPs) or a craggy nano surface is referred to as Surface-Enhanced Raman Scattering (SERS) [1,2]. In recent years, for the discovery of biomolecules and explosive molecules, SERS studies have been applicable to high-level trace elemental detection, even up to the level of single-molecule detection [3–6]. The mechanism of SERS enhancement is explained by either electromagnetic field enhancement or chemical enhancement [7–10]. Metal NPs, which possess strong localized surface plasmon resonance (LSPR) fields in the visible region of the electromagnetic spectrum, are the first choice for SERS studies [11]. Surface plasmon resonance (SPR) is a famous method whereby metal NPs display strong absorption, and they can be useful in different applications for optical field enhancement [12] and as efficient solar absorbers [13]. In recent reports, it has also been shown that graphene-based metamaterials show strong SPR absorption, and the absorption wavelength can be tuned by attaching metal NPs to graphene and be controlled with the polarization, depending on the symmetry [14,15]. Because of these attractive features, SPR has become an important feature for photonics applications, especially in

SERS. Moreover, the excitation wavelengths for SERS studies are in the visible light region. The active areas on the SERS substrate, where we observe huge enhancement in the Raman signal are referred to as "hotspots", and with a greater number of hotspots, we achieve huge enhancement in the SERS signal [5,16,17]. The size, shape, and distribution of the NPs on the substrate influence the SERS enhancement, and these have been well studied by different researchers [18–20]. The existing literature on the fabrication of SERS substrates using silver NPs is more famous due to the large SPR fields surrounding Ag nanoclusters in comparison with those of other metal NPs such as Au and Cu, but the chemical uncertainty of Ag remains a disadvantage for the use of these substrates widely in SERS studies [21]. On the other hand, Au holds decent chemical and molecular stability in comparison with Ag clusters; therefore, it is a better option for SERS studies, even though it possesses weak local electromagnetic fields with respect to Ag [22]. So, to use the contribution from the large LSPR fields of Ag and to reduce the effect of chemical instability, another alternative method is to opt for bimetallic or alloy NPs of Au and Ag with different compositions [23–25]. By varying the concentrations of both Au and Ag and adjusting the synthesis conditions, the LSPR peak of Au-Ag alloy-type composite NPs can be tuned, resulting in high SERS enhancement [26]. Therefore, there is significant potential to meet the high demand for developing low-cost substrates with consistent and high SERS enhancement factors (EFs) in relation to achievable SERS applications. Therefore, the quest for identifying low-cost substrates with unvarying and high SERS EFs is of great interest and important in relation to the SERS applications that can be achieved.

The method of synthesis of alloy NPs with tunable SPR peak positions plays a crucial role in the fabrication of SERS-related substrates. There are many reports available on the synthesis of pure-alloy-type and core-shell-type Au-Ag NPs using different methods such as chemical synthesis, the laser ablation of bulk targets in liquids, ion irradiation, etc. [27–30]. But all these methods have different issues in the fabrication process; either we end up with hazardous chemical byproducts or high-cost synthesis procedures. There are reports available on the synthesis of bimetallic NPs via the femtosecond laser ablation of bulk bimetallic targets of respective metal alloys in liquids, which result in the formation of bimetallic NPs and are used for SERS-based sensing and explosive detection [31–35]. Here, we report a simple, single-step, and cost-effective laser irradiation method for synthesizing alloy NPs by directly irradiating the metallic salt solution obtained from the precursors. We reported the synthesis of Au-Ag alloy NPs by simple, green, and single-step picosecond laser irradiation of the precursor solutions, which contain metallic slats [36]. The laser irradiation-induced heating provides the necessary energy to reduce the salts in the solution to form the required NPs in a colloidal form. These NPs are free from hazardous byproducts, pollutants, and contaminated side products. In our present study, we examined the efficacy of bimetallic NPs synthesized by laser irradiation method for the SERS enhancement of crystal violet (CV) dye molecules at various concentrations and discovered that the Au concentration in the bimetallic NPs and other synthesis conditions used to manufacture these NPs significantly enhance the SERS signal from these bimetallic NPs.

## 2. Materials and Methods

The gold (III) chloride trihydrate (HAuCl$_4$·3H$_2$O), silver nitrate (AgNO$_3$), and PVA that were procured from Aldrich and Acros Organics, respectively, possessed high purity and were used without any further purification. All the glassware was cleaned using an aqua regia solution (3:1 of HCl:HNO$_3$) and then washed in double-distilled water before using them in the synthesis procedure. The LSPR band of Au-Ag based alloy NPs was tuned by preparing different combinations of the precursor solutions of HAuCl$_4$, AgNO$_3$, and PVA, as given in Table 1. A 1%wt PVA polymer solution was used throughout the synthesis procedure for all the samples. After mixing, these solutions were stirred for 1 h continuously and were irradiated using ps laser pulses with different energies and irradiation times of the laser. For the SERS studies, we used the bimetallic NPs prepared

with an optimized energy of 140 mJ/cm$^2$ and a 30 min irradiation time. A schematic representation of the experimental setup used for ps laser irradiation is given in Figure 1.

**Table 1.** Details of the Au, Ag precursor solutions and the surface plasmon absorption wavelength for different samples.

| Au:Ag | HAuCl$_4$ (mL) | AgNO$_3$ (mL) | PVA (mL) | Total Vol (mL) | Abs. Peak (nm) |
|---|---|---|---|---|---|
| 1:0 | 4 | 0 | 1 | 5 | 530 |
| 3:1 | 3 | 1 | 1 | 5 | 494 |
| 1:1 | 2 | 2 | 1 | 5 | 468 |
| 1:3 | 1 | 3 | 1 | 5 | 428 |
| 0:1 | 0 | 4 | 1 | 5 | 406 |

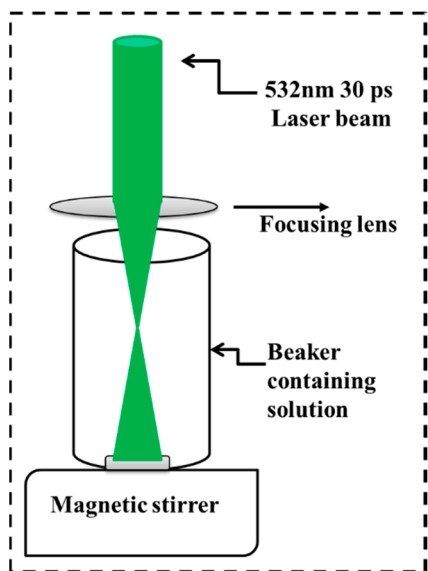

**Figure 1.** Schematic of the laser irradiation setup.

The details of the synthesis of bimetallic NPs using laser irradiation was reported previously [36,37]. In brief, we used a Q-switched Nd:YAG laser (Ekspla-2143A) of 30 ps pulse width, with a repetition rate of 10 Hz, operating at a wavelength of 532 nm to irradiate the colloidal solutions containing precursor salts. The laser fluences were optimized. and the duration of irradiation was varied from 10 min to 30 min. Conformation of the synthesis of NPs and their size distribution was obtained from TEM measurements and analysis. An FEI TECNAI G220 S-Twin model TEM instrument was used for recording the TEM data of the Au-Ag alloy NPs. The samples for TEM measurements were prepared by drop casting the respective solution of NPs on the carbon-coated copper TEM grid. A JASCO UV-Vis spectrophotometer was employed to record the optical absorption spectra of the colloidal solutions containing Au-Ag NPs. For SERS studies, the samples of Au-Ag NPs were prepared by dropping the solution of Au-Ag NPs on a Si substrate or a cover slip, and the solution was allowed to dry under normal conditions. Then, the dried substrate was used for SERS measurements for the detection of CV molecules of diverse molar concentrations prepared by sequential dilution method in water. The dried SERS substrates with Au-Ag alloy NPs were incubated with diluted CV dye solution of a fixed concentration. These samples were then air dried for about 1 h at room temperature for the uniform distribution of the CV molecules on the entire surface of the SERS substrate. A Micro-Raman spectrometer model LABRAM-HR with a 633 nm excitation wavelength was used for recording the SERS spectra of the adsorbed CV molecules on the SERS substrates. All the measurements were performed under room temperature conditions. A schematic presentation of the preparation of bimetallic NPs, the SERS substrates with the

bimetallic NPs, and the micro-Raman measurements is given in Figure 2. All the experimental characterization and the SERS measurements were performed on bimetallic NPs synthesized at an energy of 140 mJ/cm$^2$ and an irradiation time of 30 min in duration.

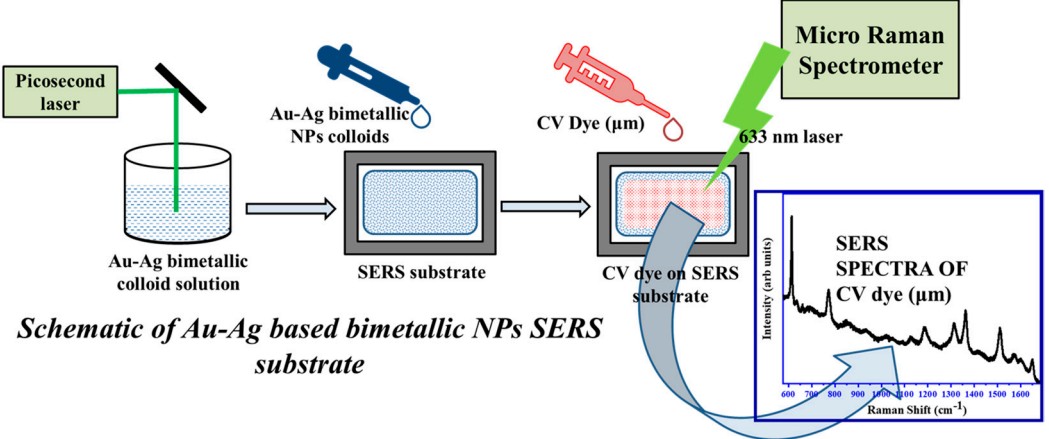

**Figure 2.** Schematic view of the preparation of SERS substrate by ps laser irradiation and the Raman measurements.

## 3. Results and Discussion

Figure 3 shows the XRD pattern of the Au-Ag bimetallic NPs with different concentration ratios. The bimetallic nature of the synthesized NPs was confirmed from the XRD patterns [38]. The XRD patterns show a bimetallic natured Au-Ag phase with the crystalline planes oriented along (1 1 1), (2 0 0), (2 2 0), and (3 1 1) directions, which match the individual phases of Au and Ag NPs [39,40]. Both the Au and Ag systems possess similar lattice constants, so they show similar peaks, and in the bimetallic phase, all the combined peaks related to both Au and Ag NPs appear. The peak observed at 2θ = 38.5° represents the (111) plane of bimetallic NPs, and it clearly indicates the NPs are highly crystalline in nature. Additionally, peaks observed at 2θ values of 44.49°, 53.2°, and 65.06° belong to (200), (020), and (220), respectively, which indicate the FCC structure of the NPs. It is also important to note that the lattice constants of the Au and Ag NPs are similar and almost match each other. In the XRD pattern, there are some additional peaks observed, which belong to silver oxide formation. These are well in agreement with the previous reports of Au-Ag bimetallic NPs [41,42].

Figure 4 shows the normalized UV-Vis absorption spectra of the colloidal Au-Ag NPs synthesized by using laser irradiation. It is clearly observed that the LSPR peak position and its full width at half maximum purely depend on the concentration ratios of Au and Ag; i.e., they, in turn, depend on the ratio of the initial weight percentages of the precursors used to obtain the solutions that are then used in the synthesis with laser irradiation. It is observed that the LSPR peak was red-shifted with the increase in the weight percentage of the Au. For pure Ag NPs, the LSPR peak was at 407 nm [43], which was shifted to 530 nm for the pure Au NPs [44]. As the ratio of Au-Ag varied, the peak position also shifted between 407 nm and 530 nm. This confirms the formation of the Au-Ag bimetallic phase in the NPs formed by a single step, clean, and green method of laser irradiation. Here, it is also observed that the LSPR peak position is more dependent on the Au-Ag concentration ratios alone and in some cases also on the average size of the individual NPs. The NPs are observed to be highly stable, as they are embedded in a PVA matrix, which prevents the NPs' direct exposure to the atmosphere and keeps the NPs free from oxidation and other effects. The UV-Vis absorption spectra shows no double peak formation; therefore, the NPs belong to a mixed alloy phase of Au and Ag, and the spectra does not show any signature of the individual NPs' presence in the alloy NPs.

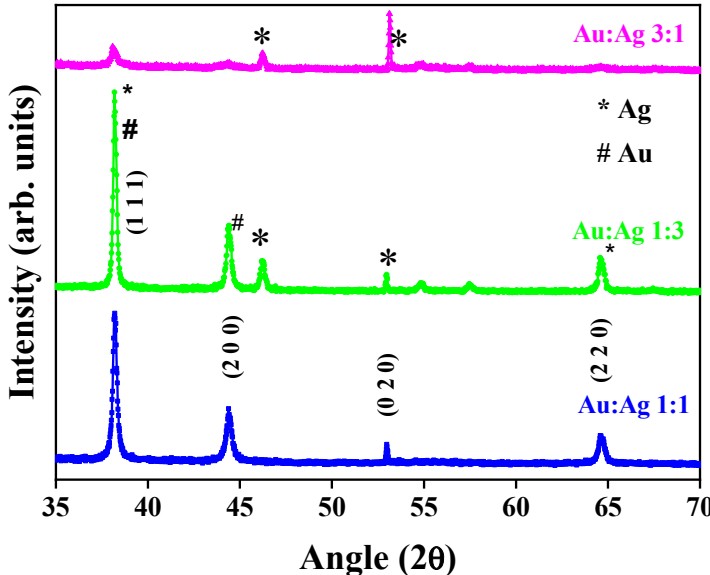

**Figure 3.** XRD Pattern of the Au-Ag alloy NPs synthesized by picosecond laser ablation of the respective salts of different ratios at a fixed energy of the laser irradiation.

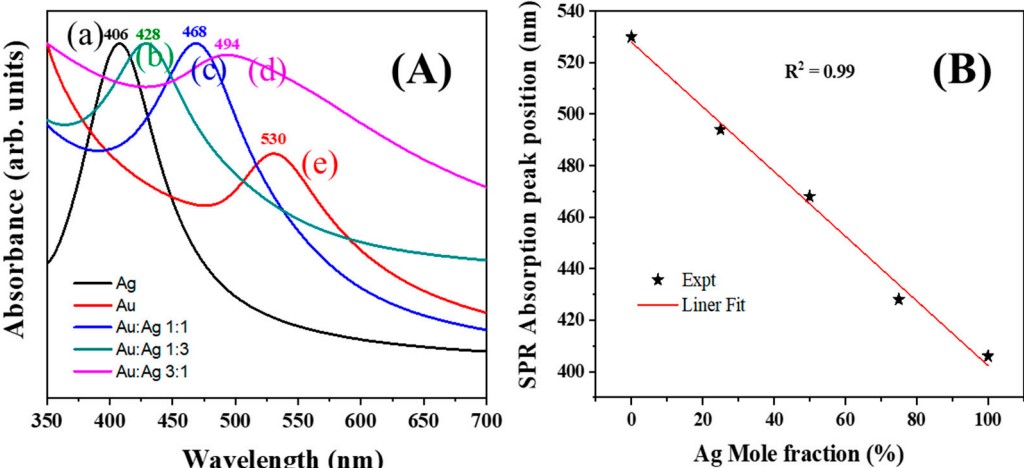

**Figure 4.** (**A**) UV−Vis absorption spectra of Au-Ag bimetallic NP colloid solutions at different concentration ratios of Au and Ag. (a) Pure Ag, (b) Au:Ag = 1:3, (c) Au:Ag = 1:1, (d) Au:Ag = 3:1, and (e) pure Au prepared by laser irradiation of the precursor solutions. (**B**) LSPR absorption peak position variation with the mole fraction of Ag in the Au-Ag bimetallic NPs.

The colloidal dispersions of the Au-Ag NPs are analyzed using TEM to determine their average size, distribution, and the morphology of the NPs. The solutions were drop casted onto carbon-coated TEM grids and used for TEM measurements after drying for a few hours. It is observed that the NPs are clearly dispersed and highly crystalline in orientation, which were confirmed from the high-resolution TEM images of the NPs.

Figure 5 presents the TEM images of the bimetallic NPs, which show the NPs are highly crystalline in nature and dispersed well in the colloidal solution. This type of NP is produced by irradiating metal salts of both Au and Ag with highly intense ps laser pulses, resulting in the reduction of the precursors and their subsequent formation into bimetallic NPs. There may be an efficient race in the process of reduction, in which the bimetallic NPs are achieved in a very short duration of the irradiation, and with a further increase in irradiation time, the growth of NPs and other processes such as alloying core-shell type formation may be achieved, which also strongly depend on the laser energy, fluence, irradiation time, and the concentration of the initial precursor salts [45]. The continuous

laser irradiation reduces the chances of the segregation of NPs and forms stable and not agglomerated NPs. These bimetallic NPs show an alloy-type nature due to the similar atomic sizes and the crystalline nature of both the Au and Ag atoms. It is observed from the high-resolution images [Figure 5C,F,I] of the bimetallic NPs that there is a difference in the lattice parameter values of the single NP, which is a clear indication that the rates of diffusion of two kinds of atoms are different. Hence, during the diffusion of Au and Ag atoms, the chance of bimetallic NPs formation is higher, and it is observed to be more prominent in larger NPs, where more atoms are involved in the formation of bimetallic NPs. This type of effect is famously known as the Kirkendall effect [46], which is a commonly observed phenomenon in the case of the synthesis of bimetallic or alloy-type systems. The high-resolution images presented in the inset of Figure 5C indicate the lattice parameter of the NPs is 0.23 nm, and it is matches with both Au and Ag. Similarly, the image presented in Figure 5F also indicates bimetallic related lattice parameters, which clearly indicate the NP is a combination of both Au and Ag, i.e., bimetallic. This analysis complements the observations from the UV-Vis absorption spectra. The shift in the LSPR peak position confirms the concentration of the Ag/Au in the composite Au-Ag NPs is related to the ratio of the Au and Ag in the bimetallic NPs by shifting the peak position accordingly. The Au-AgNPs and their LSPR peak position can be tuned by changing the Au concentration in between the SPR peak of silver (406 nm) and the pure SPR peak of gold (536 nm). If the variation in the composition of Ag/Au is not gradual, this kind of fine tuning is impossible. Here, in all the samples the peak position can shift from 406 to 530 nm, and it clearly indicates the NPs are only bimetallic, and the Au concentration shifts the LSPR peak position. On the other hand, the EM results show the presence of individual single NPs with both Au- and Ag-related lattice parameters. Hence, both these results together indicate the formation of bimetallic Au-Ag NPs as a result of laser irradiation.

The average size of bimetallic NPs is estimated, and Figure 6A–C represent the average size of the bimetallic NPs formed with Au:Ag 3:1, 1:1, 1:3 molar ratios. The bimetallic NPs vary in size; with the increase in the concentration of Ag, the average size changes from 32 nm to 24 nm. The selected area electron diffraction (SAED) patterns show that these bimetallic NPs are highly crystalline. All the crystalline planes of the bimetallic NPs such as (111), (200), and (220) are observed and marked at their respective positions in SAED pattern. The LSPR absorption peak of the bimetallic NPs can be tuned with the laser energy, irradiation time, and the concentration of the initial precursor solutions. Therefore, the absorption peak of LSPR can be tuned with changing laser irradiation parameters, and it is given for different laser parameters for a fixed concentration of Au-Ag in Table 2. It is observed that the LSPR absorption wavelength changes with the irradiation time at a fixed energy of ps laser pulses. Also, the LSPR peak position can be tuned with the laser energy for a fixed duration of irradiation. The increase in the irradiation time decreases the LSPR peak position and shifts to the lower wavelength side. It is also observed that the LSPR wavelength can also be tuned by changing the Ag mole fraction in bimetallic Au-Ag systems. Hence, it is understood that by simply playing with the laser irradiation parameters and the concentration of the Ag salts in the Au-Ag based bimetallic NPs, the LSPR absorption wavelength can be tuned between the individual absorption wavelengths of Au and Ag. It is also observed that when pure Au and Ag NP colloids are mixed with each other the LSPR absorption spectra shows the presence of double peaks, which belongs to both Au and Ag NPs. In the case of bimetallic NPs, only a single peak is observed, and no trace of a double peak is observed.

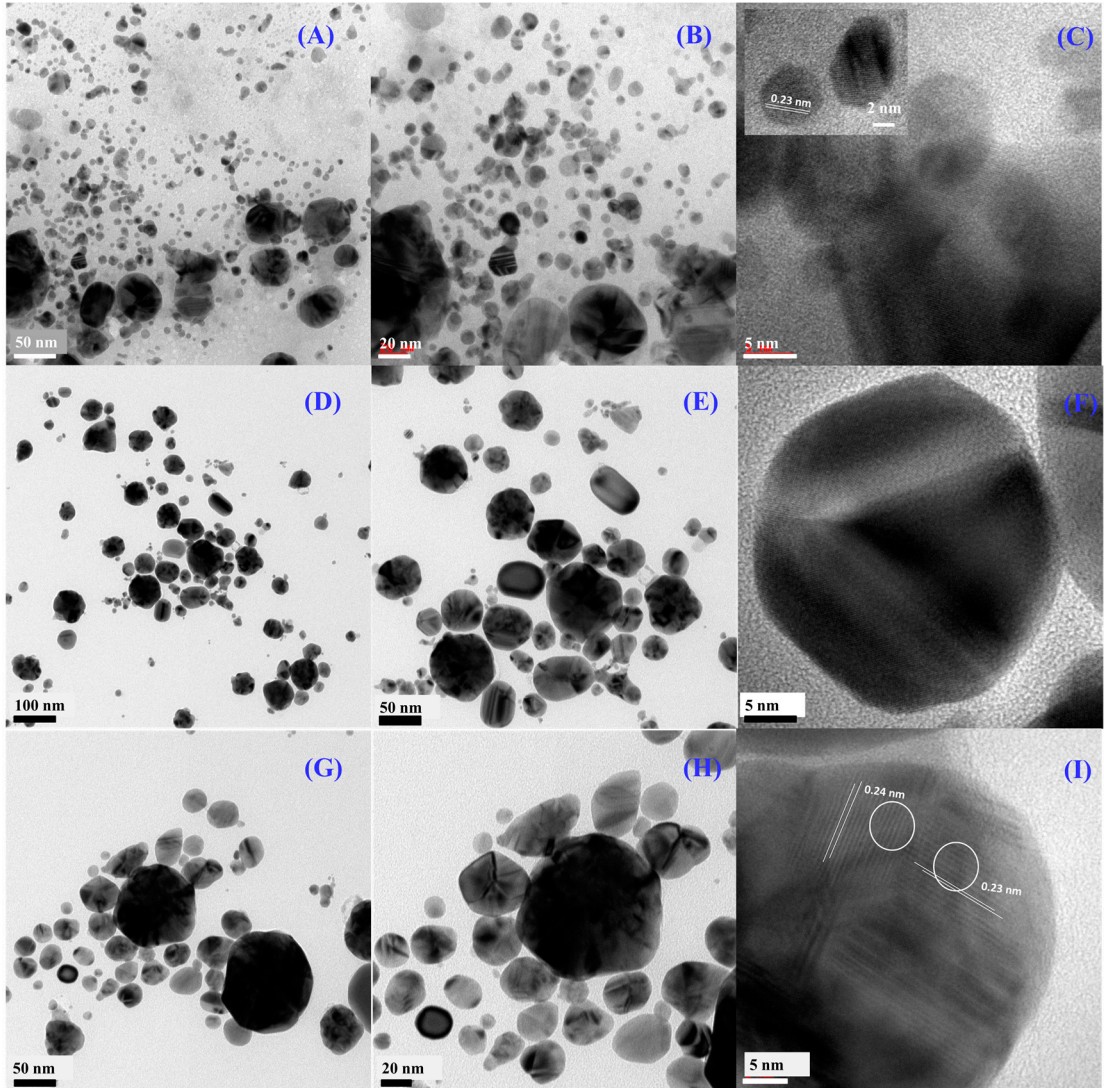

**Figure 5.** TEM images of the Au-Ag bimetallic NPs in different magnifications. (**A–C**) are for the 3:1, (**D–F**) are for the 1:1, and (**G–I**) are for the 1:3 Au-Ag ratio of the precursor solutions. The inset of (**C**) shows the respective high-resolution image. The lattice parameter values are marked for reference on the images.

- SERS studies of bimetallic NPs

Figure 7 represents the SERS spectra of the 1μM CV dye deposited on the bimetallic NPs dropped on Si substrates. These SERS spectra are the averaged spectra from 10 different spots on the bimetallic NP substrate. The colloidal NP solution was deposited by drop casting on the Si substrates and then air dried for a few hours. The CV dye solution with the desired concentration of 1 μM is dropped on the NP film on SERS substrates, and we recorded the SERS spectra. We have observed that the bimetallic NPs can be used as efficient SERS platform for the detection of dye molecules at the lowest concentration. The different Au concentration samples are used for SERS studies. Here, the important observation is that the presence of Au in bimetallic alloy NPs influences the SERS activity. In general, the Ag NPs show strong oxidation effects, and here, the addition of Au into Ag produces Au-Ag bimetallic NPs, and the oxidation effects are reduced due to Au.

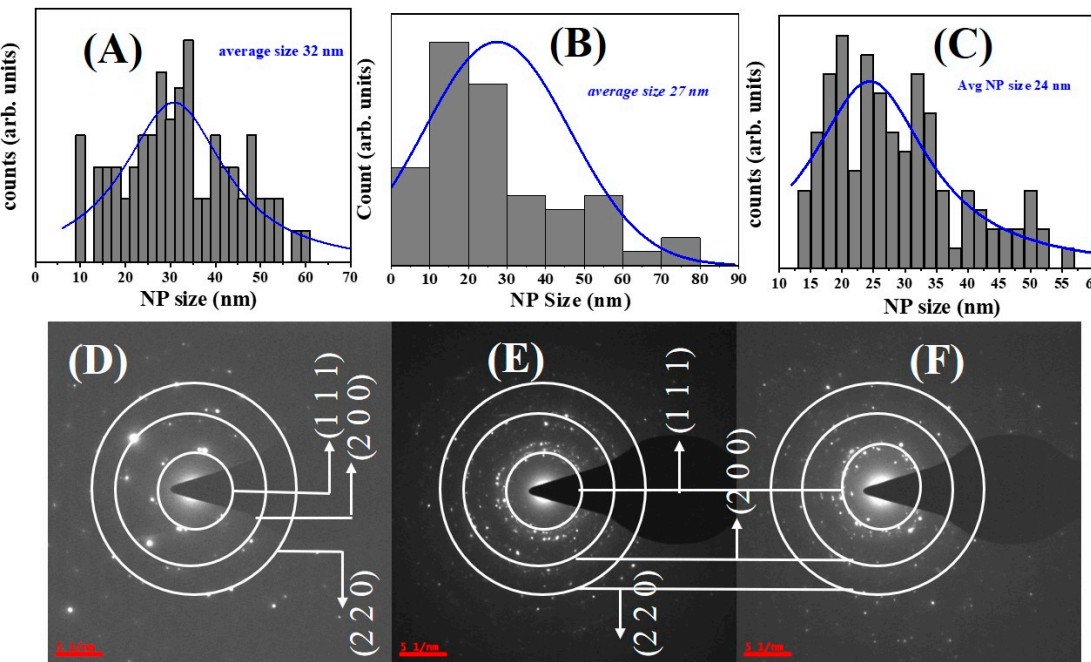

**Figure 6.** (**A–C**) are the particle size distribution histograms of the bimetallic NPs, and (**D–F**) represent the selected area electron diffraction patterns of the Au-Ag bimetallic NPs.

**Table 2.** Variation in the LSPR peak position of the Au-Ag (1:1) sample for different laser irradiation parameters.

| S. No. | Energy or Fluence (mJ/cm$^2$) | LSPR Abs Peak Position (nm) | Irradiation Time (min) |
|---|---|---|---|
| 1 | 185 | 470 | 10 |
| 2 | 185 | 462 | 20 |
| 3 | 185 | 458 | 30 |
| 4 | 140 | 478 | 20 |
| 5 | 140 | 468 | 30 |
| 6 | 100 | 496 | 20 |

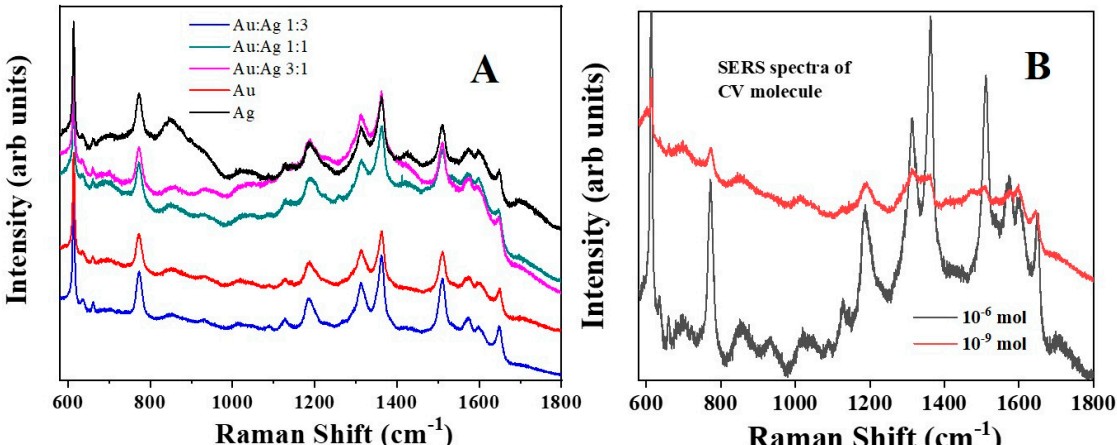

**Figure 7.** (**A**) SERS spectra of the 1 μM dye dropped on the bimetallic NPs prepared by laser ablation and deposited on Si substrates. (**B**) shows the comparative SERS spectra of the μM and nM dye on the Au:Ag = 1:1 bimetallic NP substrate.

The Au-Ag bimetallic NPs and the CV dye leads to the enhancement of all the predominant peaks of the dye molecules attached to Au-Ag NPs. It is observed that even at

the low dye concentration the Raman signals are highly enhanced. The SERS analytical enhancement factor (*AEF*) from these bimetallic Au-Ag NPs substrates has been estimated by using the analytical formula as given in Equation (1) [47],

$$AEF = \frac{\left(\frac{I_{SERS}}{C_{SERS}}\right)}{\left(\frac{I_{RS}}{C_{RS}}\right)} \tag{1}$$

where $I_{SERS}$ represents the Raman signal intensity obtained for the SERS substrate under a certain concentration of $C_{SERS}$, and $I_{RS}$ corresponds to the Raman signal intensity obtained under non-SERS conditions at a concentration of $C_{RS}$. The general vibrational bands of the CV dye fall the within the wavenumber range of 600 to 1800 cm$^{-1}$ of the Raman spectra. The most prominent vibrational peak positions are the C-H in-plane ring bending vibrations of the CV dye molecule present at wavenumber ~1170–1190 cm$^{-1}$, this is taken as the reference [48], and the respective Raman intensities for different samples are used for calculating the *AEF*. All the *AEFs* are the averages of the measurements performed from different locations on the SERS substrates chosen randomly. The estimated *AEF* values are of the order ~10$^8$. The vibrational Raman modes of the CV dye molecule are assigned by following the previous reports, and the details are given in Table 3 [49]. Figure 7A represents the SERS spectra of the 1 μM CV dye dropped on the bimetallic NPs prepared by laser irradiation and deposited on Si substrates with different Au-Ag molar ratios. Figure 7B shows the comparative SERS spectra of the μM and nM dye on the Au:Ag = 1:1 bimetallic NPs substrate. This clearly indicates that the SERS substrates with bimetallic NPs are able to detect even very low concentrations of the dye molecules present on the substrates.

**Table 3.** The Raman vibrational peak positions observed for the CV dye molecule adsorbed on the bimetallic NP substrates.

| S. No. | Raman Peak Position Wavenumber (cm$^{-1}$) | Peak Position Assign | *AEF* on Au:Ag = 1:1 Substrate |
|---|---|---|---|
| 1 | 500–560, 615, 690–986 | The aromatic ring skeletal vibrations | $1.6 \times 10^7$ |
| 2 | 726–795, 817 | C-H out-of-plane ring bending vibrations | $2.4 \times 10^7$ |
| 3 | 1170–1190 | C-H in-plane ring bending vibrations | $1.2 \times 10^8$ |
| 4 | 1360–1390 | N-phenyl stretching vibrations | $8.3 \times 10^7$ |
| 5 | 1310, 1444 | The aromatic ring C-C stretching vibrations | $6.2 \times 10^7$ |
| 6 | 1538–1584, 1620 | The aromatic ring C-C stretching vibrations | $9.2 \times 10^7$ |

## 4. Conclusions

We presented a simple fabrication method for the preparation of Au-Ag bimetallic NPs with a tunable LSPR wavelength using picosecond laser irradiation. The LSPR wavelength of the Au-Ag bimetallic NPs is tuned with Au-Ag molar concentrations at fixed laser irradiation parameters. The as-prepared bimetallic NPs are used for the SERS studies with CV dye molecules of different concentrations. The average EFs display variation depending on the synthesis conditions of the Au-Ag bimetallic alloy NPs and the Au concentration. The CV dye samples showed EFs as high as of the order of 10$^8$, and we were able to detect even nmol of the sample on these Au-Ag alloy NPs. These studies demonstrate a simple, single step, and green method of fabrication of Au-Ag alloy-based nanocomposites that can be suitable for SERS investigations and the usage of bimetallic Au-Ag NPs to tune for high EFs.

**Author Contributions:** P.B.: Methodology, Investigation, Formal analysis. M.A.T.: Investigation, Formal analysis. M.H.D.: Investigation, Writing—original draft, Validation. D.N.R.: Resources, Writing—original draft, Validation. P.G.K.: Formal analysis, Writing—original draft. V.S.: Conceptualization, Methodology, Formal analysis, Validation, Resources, Funding acquisition, Writing—original draft, Supervision. All authors have read and agreed to the published version of the manuscript.

**Funding:** The research received funding from the SERB International Research Experience (SIRE) fellowship with grant SIR/2022/000618 and UGC-DAE-CSR-Indore through a Collaborative Research Scheme (CRS) project number CRS/2022-23/01/687.

**Data Availability Statement:** Data will be made available on request to the corresponding author.

**Acknowledgments:** V.S. acknowledges the financial support from SERB-India for awarding the SERB International Research Experience (SIRE) fellowship with grant SIR/2022/000618 and S. Amuruso for hosting in his group at the University of Naples Federico II, Napoli, Italy. The authors acknowledge the financial support from UGC-DAE-CSR-Indore through a Collaborative Research Scheme (CRS) project number CRS/2022-23/01/687. We are thankful to the UGC NRC Centre of the School of Physics and the Centre for Nanotechnology at UOHYD for extending experimental facilities, such as the Raman, FESEM, PL, TEM, and XRD facilities.

**Conflicts of Interest:** The authors declare no conflict of interest.

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
