# Peer review of "Surface-Enhanced Raman Scattering Studies of Au-Ag Bimetallic Nanoparticles with a Tunable Surface Plasmon Resonance Wavelength Synthesized by Picosecond Laser Irradiation"

_photonics, doi:10.3390/photonics10121345_

Round 1

Reviewer 1 Report

Comments and Suggestions for Authors

The manuscript presented the synthesis method of a nanoparticle system whose SERS wavelength is controlled by ps laser irradiation parameters and the materials. The results are interesting to the broad audience, and the results can have SERS based sensing technology. However, I'd recommend some further revisions to the authors to better meet the standard of publication.

1. Figure 3: It'd be better to place the legend just next to each line for easier reading. It is also better to use the same color code for different Au:Ag ratios as Figure 4A.

2. Figure 3: Could the authors list the material and the orientation of each peak? Does the orientation match TEM results in Figure 5?

3. Could the authors elaborate more on what the audience would observe from Figure 7A? This seems unclear. Also, would substring the background of each spectra help the data presentation?

4. Format issue: The authors used Figure 1/2/3, but Fig. 4/5/6/7.

Author Response

We thank the reviewers for spending their time on our manuscript and to spend valuable suggestions to improve the strength of the manuscript. The queries raised by the reviewers have improved the quality of the manuscript to a great extent. We tried to respond to the queries within the limited frame of our understanding. Every query is addressed and incorporated in the revised manuscript without losing the flow of the main text. Point-by-point responses to the queries are appended below…

If the reviewers found any query unaddressed/wrongly addressed, please let us know to revise the manuscript accordingly.

The manuscript presented the synthesis method of a nanoparticle system whose SERS wavelength is controlled by ps laser irradiation parameters and the materials. The results are interesting to the broad audience, and the results can have SERS based sensing technology. However, I'd recommend some further revisions to the authors to better meet the standard of publication.

Comment1: Figure 3: Placing the legend next to each line for easier reading would be better. It is also better to use the same color code for different Au: Ag ratios as in Figure 4A.

Response: The modified colour code has been incorporated in the revised manuscript.

Comment2: Figure 3: Could the authors list each peak's material and orientation? Does the orientation match the TEM results in Figure 5?

Response: We have identified and marked the orientations in the figure and these results agree with TEM. Our analysis confirms that the structure of the synthesized Au-Ag bimetallic nanomaterial has the crystal phase of Face Cantered Cube (FCC).

Comment3: Could the authors elaborate more on what the audience would observe from Figure 7A? This seems unclear. Also, would substring the background of each spectra help the data presentation?

Response: We modified the figure and elaborated in detail in the revised manuscript.

Comment4: Format issue: The authors used Figure 1/2/3, but Fig. 4/5/6/7.

Response: We are very sorry to say that, we did not understand this query. We wonder to know whether reviewer is asking about the citation of the figure numbers in the manuscript wrongly? Anyways, we cross checked the figures one more time and as per our best of the knowledge they are placed and cited properly.

Reviewer 2 Report

Comments and Suggestions for Authors

In this article, the authors propose a simple and green method for preparing Au Ag bimetallic nanoparticles (NP) with tunable surface plasmon resonance (SPR) wavelength using picosecond laser irradiation. The Au-Ag alloy nanoparticles are produced by irradiating the metallic salts with a picosecond laser in a polyvinyl alcohol (PVA) matrix in a single step process. The SPR wavelength of the Au-Ag bimetallic NPs is observed to be shift/changed with the Au-Ag concentration and the laser irradiation parameters. The Au-Ag NPs embedded in the PVA matrix are observed to be gifted for their use in surface-enhanced Raman scattering (SERS) applications. These studies demonstrate a simple, single-step, and green method of fabrication of an Au-Ag alloy-based nanocomposites that can be suitable for SERS investigations and the usage of the bimetallic Au-Ag NP systems to tune for the high EF. I believe that publication of the manuscript may be considered only after the following issues have been resolved.

1.    Is the Raman enhancement performance of this SERS substrate stable over a large area? The author needs to provide corresponding representations.

2.    What is the physical mechanism behind the superior performance of this composite system?

3.    In Figure 3, the corresponding peak positions need to be annotated by the author in the figure.

4.    The introduction can be improved. The articles related to some applications of surface-plasmon resonance should be added such as

5.    Micromachines 2023, 14, 953; Optics Express, 30(20), 35554-35566, 2022; Applied Thermal Engineering 230 (2023) 120841; IEEE Photonics Technology Letters, vol.29(3), 295-298, 2017.

6.    The English expression of the whole article needs to be further improved.

Comments on the Quality of English Language

Minor editing of English language required

Author Response

In this article, the authors propose a simple and green method for preparing Au Ag bimetallic nanoparticles (NP) with tunable surface plasmon resonance (SPR) wavelength using picosecond laser irradiation. The Au-Ag alloy nanoparticles are produced by irradiating the metallic salts with a picosecond laser in a polyvinyl alcohol (PVA) matrix in a single step process. The SPR wavelength of the Au-Ag bimetallic NPs is observed to be shift/changed with the Au-Ag concentration and the laser irradiation parameters. The Au-Ag NPs embedded in the PVA matrix are observed to be gifted for their use in surface-enhanced Raman scattering (SERS) applications. These studies demonstrate a simple, single-step, and green method of fabrication of an Au-Ag alloy-based nanocomposites that can be suitable for SERS investigations and the usage of the bimetallic Au-Ag NP systems to tune for the high EF. I believe that publication of the manuscript may be considered only after the following issues have been resolved.

Comment1: Is the Raman enhancement performance of this SERS substrate stable over a large area? The author needs to provide corresponding representations.

Response: We thank the reviewer for providing the significant queries. The SERS active substrates were prepared by the drop cast method and all these substrates were tested for their uniform dispersity of the Ag-Au nanoparticles. However, these initial investigations were performed in trial and error to quantify the volume of the colloidal dispersions to be utilized to drop cast onto the Si substrate. Finally, we could fix the volume of the colloidal dispersion to 20 μL (measured with a micro-pipet) to grow a SERS active substrate with uniform dispersity on the Si substrate. In this way, we prepared n number of SERS active platforms in the first step. Latron, we checked the Raman signature enhancements of the analytes at low concentrations from the SERS active substrates from different locations and confirmed the uniform elevation of the Raman signals. However, we took this second step to confirm the uniform elevation of Raman signals from different locations. Thus, we could conclude that the performance of our bi-metallic SERS active platforms are said to be stable over a large area to produce consistent Raman signature enhancements.

YES, the Raman recorded at different positions is averaged and similar over a large area. Now we have added additional information in the discussion.

Comment2: What is the physical mechanism behind the superior performance of this composite system?

Response: Plasmonic metals particularly gold, copper, and silver have a complex dielectric function in which the real part is more negative, responsible for the induction of the fields in them, and the imaginary part is less positive but very significant in demonstrating the absorption of optical fields. This peculiar behavior of plasmonic metals facilitates them to create evanescent fields in their vicinity due to the excitation of surface plasmon resonances (SPRs). In view of this, gold/silver/copper are having their individual strengths and weaknesses. In particular silver nanomaterials can produce more local fields but silver is more prone to oxidation. On the other hand, gold produces moderate, consistent less evanescent fields compared to silver. If these two metals are mixed, one can tune the SPR wavelength and the composite system overcomes oxidation. The bi-metallic SERS active platforms of silver and gold produce a very strong SPR and, hence, the local fields. The fraction of silver/gold in the composite Ag-Au nanoparticle modifies the SPR frequency which facilitates the coupling of incident photons to the oscillating Ag-Au nanoparticle dipoles. Consequently, these composites of Ag-Au results in the gigantic enhancement of the Raman signatures of analyte molecules adsorbed on them.

The physical mechanism is … The revised manuscript we have added additional explanations too.

Comment3: In Figure 3, the corresponding peak positions need to be annotated by the author in the figure.

Response: We have included them in the changes done in revised manuscript.

Comment4: The introduction can be improved. The articles related to some applications of surface-plasmon resonance should be added such as Micromachines 2023, 14, 953; Optics Express, 30(20), 35554-35566, 2022; Applied Thermal Engineering 230 (2023) 120841; IEEE Photonics Technology Letters, vol.29(3), 295-298, 2017.

Response: The introduction is modified according to the suggestions given by the reviewer and the discussion is added in the revised manuscript along with the citing of the above-mentioned articles without losing the flow text/content.

Comment5: The English expression of the whole article needs to be further improved.

Response: The English is rechecked and verified for grammatical and other errors in the revised manuscript.

Reviewer 3 Report

Comments and Suggestions for Authors

The article is devoted to the green method for preparing Au-Ag bimetallic nanoparticles with tunable surface plasmon resonance (SPR) wavelength using picosecond laser irradiation. This topic is relevant from a practical point of view. At the same time, the use of Ag, Au, and Cu metal nanoparticles has been discussed many times in the literature, including the use of bimetallic nanoparticles. The authors use bimetallic Au-Ag NPs embedded in the PVA matrix. And they study the effect of these particles on surface-enhanced Raman scattering (SERS) depending on the ratio of Au and Ag in the alloy. At the same time, the manuscript contains a number of shortcomings that must be corrected before accepting the article for publication.

1) In Fig. 3, it is necessary to indicate the planes, as well as supplement the X-ray pattern with data on Au and Ag for comparative analysis.

2) Table 1 shows the peak surface plasmon absorption wavelength for different samples of 407 nm and 428 nm for Au:Ag ratios of 0:1 and 1:3, respectively. And in Fig. 6, the wavelengths of 406 nm and 430 nm are indicated. It is necessary to give explanations in the form of an error of the device or something else.

3) The figure caption to Fig. 4 contains the decoding (a)-(e), which are not in the figure.

4) In fig. 5 and 6, the color and font size of the scalbar caption are poorly chosen, which makes the scale bar hard to read.

5) Figure 5 should show interplanar distances in HRTEM images corresponding to Au and Ag to demonstrate the high crystallinity of the nanoparticles.

6) Line 159-160. Authors speak about the high crystallinity of bimetallic nanoparticles. In this case, crystallinity is observed only in 1:3 Au:Ag (Fig. 5, (I)). Then there are explanations about hollow type structures in the NPs. This causes conflicting opinions.

7) Line 240. Written 108, probably wanted to write 108.

8) Figure 6 (D)-(F) shows diffraction patterns that correspond to different sizes of bimetallic nanoparticles. For a clear understanding, it is necessary to apply diffraction rings corresponding to Au and Ag and mark them in the figure. In general, the size of the diffraction rings does not change depending on the size of the nanoparticles, because the interplanar distances remain the same, corresponding to Au and Ag. But the authors demonstrated different diffraction patterns.

9) To confirm Au:Ag ratios of 1:1, 1:3, 3:1, XPS or EDX data must be provided.

10) The Conclusions section says "…simple fabrication method for the preparation of Au-Ag bimetallic nanoparticles (NPs) with tunable surface plasmon resonance (SPR) wavelength by using picosecond laser irradiation. The SPR wavelength of the Au-Ag bimetallic NPs can be tuned with Au-Ag molar concentrations and the laser irradiation parameters. The average enhancement factors (EF) show variation depending on the size and synthesis conditions of the Au-Ag bimetallic alloy NPs and the Au concentration". There are no data on the effect of the size of Au-Ag bimetallic nanoparticles on absorption spectra and SERS spectra.

Author Response

The article is devoted to the green method for preparing Au-Ag bimetallic nanoparticles with tunable surface plasmon resonance (SPR) wavelength using picosecond laser irradiation. This topic is relevant from a practical point of view. At the same time, the use of Ag, Au, and Cu metal nanoparticles has been discussed many times in the literature, including the use of bimetallic nanoparticles. The authors use bimetallic Au-Ag NPs embedded in the PVA matrix. And they study the effect of these particles on surface-enhanced Raman scattering (SERS) depending on the ratio of Au and Ag in the alloy. At the same time, the manuscript contains a number of shortcomings that must be corrected before accepting the article for publication.

  1. Comment: In Fig. 3, it is necessary to indicate the planes, as well as supplement the X-ray pattern with data on Au and Ag for comparative analysis.

Response: The planes and respective peaks are indicated and also discussion part is modified.

  1. Comment: Table 1 shows the peak surface plasmon absorption wavelength for different samples of 407 nm and 428 nm for Au:Ag ratios of 0:1 and 1:3, respectively. And in Fig. 6, the wavelengths of 406 nm and 430 nm are indicated. It is necessary to give explanations in the form of an error of the device or something else.

Response: The error bars are included and explanations are changed in revised version.

  1. Comment: The figure caption to Fig. 4 contains the decoding (a)-(e), which are not in the figure.

Response: The figure is changed as per the caption.

  1. Comment: In fig. 5 and 6, the color and font size of the scale bar caption are poorly chosen, which makes the scale bar hard to read.

Response: The scalebar and fronts in the figure are changed to white colour from red for the clear visibility.

  1. Comment: Figure 5 should show interplanar distances in HRTEM images corresponding to Au and Ag to demonstrate the high crystallinity of the nanoparticles.

Response: As suggested by reviewer The interplanar distances are marked and they are matching with Au-Ag bimetallic NPs.

  1. Comment: Line 159-160. Authors speak about the high crystallinity of bimetallic nanoparticles. In this case, crystallinity is observed only in 1:3 Au:Ag (Fig. 5, (I)). Then there are explanations about hollow type structures in the NPs. This causes conflicting opinions.

Response: The explanations related to hollow type structures is removed to avoid conflicting opinions. The crystalline nature is observed in all the NPs. The images are now indicated with crystalline planes.

  1. Comment: Line 240. Written 108, probably wanted to write 108.

Response: The respective changes are made in the manuscript revision.  

  1. Comment: Figure 6 (D)-(F) shows diffraction patterns that correspond to different sizes of bimetallic nanoparticles. For a clear understanding, it is necessary to apply diffraction rings corresponding to Au and Ag and mark them in the figure. In general, the size of the diffraction rings does not change depending on the size of the nanoparticles, because the interplanar distances remain the same, corresponding to Au and Ag. But the authors demonstrated different diffraction patterns.

Response: Diffraction rings are applied in revision. The interplanar distances are same and demonstration is only due to different concentrations of Au and Ag also both Au and Ag have almost same lattice parameters.

  1. Comment: To confirm Au:Ag ratios of 1:1, 1:3, 3:1, XPS or EDX data must be provided.

Response: We agree with the reviewer, particularly regarding this comment to have EDAX data to confirm the Ag/Au concentration ratio in the Ag-Au nanoparticles. However, EDAX can't give an overall confirmation about the compositions since EDAX only confines to a very small area of the nano aggregate. On the other hand, it gives average information in the considered EDAX window but not in individual particles. UV-vis absorption peak shift confirms the concentration of the Ag/Au in the composite Ag-Au nanoparticles by shifting the peak position accordingly. The Ag-Au NPs and their peak position is tuning as a function of Au concentration in between the pure plasmon resonance peak of silver (430 nm) and the pure plasmon peak position of gold (530 nm). If the variation in the composition of Ag/Au is not gradual this kind of tuning is impossible. Here in all the samples the peak position was able to shift from 406 to 530 nm and it clearly indicates the NPs are bimetallic and the Au concentration shift the SPR resonance peak.

  1. Comment: The Conclusions section says "…simple fabrication method for the preparation of Au-Ag bimetallic nanoparticles (NPs) with tunable surface plasmon resonance (SPR) wavelength by using picosecond laser irradiation. The SPR wavelength of the Au-Ag bimetallic NPs can be tuned with Au-Ag molar concentrations and the laser irradiation parameters. The average enhancement factors (EF) show variation depending on the size and synthesis conditions of the Au-Ag bimetallic alloy NPs and the Au concentration". There are no data on the effect of the size of Au-Ag bimetallic nanoparticles on absorption spectra and SERS spectra.

Response: The conclusion section is modified and other NPs with different tunable SPR resonance peak with different amounts of Au and Ag precursor salts was reported earlier.

Round 2

Reviewer 1 Report

Comments and Suggestions for Authors

The authors addressed some of my concerns, however other concerns need to be further addressed to meet the standard of publications. The rest of the concerns are more data presentation and format related.

1. Figure 7A - I still think it is better to subtract the background for better comparison among different spectra.

2. The authors used Figure 1-3 and Fig. 4-7. "Figure" or "Fig." should be consistent through the manuscript.

Author Response

Dear Reviewer 

We thank you for your patience and suggestions for the improvement of our manuscript. We have modified the manuscript and figures and captions now in the revised submission. 

Reviewer 2 Report

Comments and Suggestions for Authors

Accept in present form

Author Response

Thank you for your time and suggestions for improving our manuscript

Reviewer 3 Report

Comments and Suggestions for Authors

The authors responded to all comments.

Author Response

Thank you for your time and suggestions